# Excess mortality attributable to antimicrobial-resistant bacterial bloodstream infection at a tertiary-care hospital in Indonesia

Patricia M. Tauran[1]*, Irawaty Djaharuddin[2,3], Uleng Bahrun[2,4], Asvin Nurulita[2,4], Sudirman Katu[2,5], Faisal Muchtar[2,6], Ninny Meutia Pelupessy[2,7], Raph L. Hamers[8,9,10], Niholas P. J. Day[1,9], Mansyur Arif[2,4], Direk Limmathurotsakul[1,9,11]

1 Mahidol Oxford Tropical Medicine Research Unit, Faculty of Tropical Medicine, Mahidol University, Bangkok, Thailand, 2 Dr. Wahidin Sudirohusodo Hospital, Makassar, South Sulawesi, Indonesia, 3 Department of Pulmonology and Respiratory Medicine, Faculty of Medicine, Hasanuddin University, Makassar, South Sulawesi, Indonesia, 4 Department of Clinical Pathology, Faculty of Medicine, Hasanuddin University, Makassar, South Sulawesi, Indonesia, 5 Department of Internal Medicine, Faculty of Medicine, Hasanuddin University, Makassar, South Sulawesi, Indonesia, 6 Department of Anesthesiology, Faculty of Medicine, Hasanuddin University, Makassar, South Sulawesi, Indonesia, 7 Department of Pediatrics, Faculty of Medicine, Hasanuddin University, Makassar, South Sulawesi, Indonesia, 8 Eijkman-Oxford Clinical Research Unit, Jakarta, Indonesia, 9 Centre for Tropical Medicine and Global Health, Nuffield Department of Medicine, University of Oxford, Oxford, United Kingdom, 10 Faculty of Medicine, Universitas Indonesia, Jakarta, Indonesia, 11 Department of Tropical Hygiene, Faculty of Tropical Medicine, Mahidol University, Bangkok, Thailand

* patricia@tropmedres.ac

**Data Availability Statement:** Data available in article supporting information.

## Abstract

The burden of antimicrobial-resistant (AMR) infections in low and middle-income countries (LMICs) is largely unknown. Here, we evaluate attributable mortality of AMR infections in Indonesia. We used routine databases of the microbiology laboratory and hospital admission at Dr. Wahidin Sudirohusodo Hospital, a tertiary-care hospital in South Sulawesi from 2015 to 2018. Of 77,752 hospitalized patients, 8,341 (10.7%) had at least one blood culture taken. Among patients with bacteriologically confirmed bloodstream infections (BSI), the proportions of patients with AMR BSI were 78% (81/104) for third-generation cephalosporin-resistant (3GCR) *Escherichia coli*, 4% (4/104) for 3GCR plus carbapenem-resistant *E. coli*, 56% (96/171) for 3GCR *Klebsiella pneumoniae*, 25% (43/171) for 3GCR plus carbapenem-resistant *K. pneumoniae*, 51% (124/245) for methicillin-resistant *Staphylococcus aureus*, 48% (82/171) for carbapenem-resistant *Acinetobacter* spp., and 19% (13/68) for carbapenem-resistant *Pseudomonas aeruginosa*. Observed in-hospital mortality of patients with AMR BSI was 49.7% (220/443). Compared with patients with antimicrobial-susceptible BSI and adjusted for potential confounders, the excess mortality attributable to AMR BSI was -0.01 (95% CI: -15.4, 15.4) percentage points. Compared with patients without a BSI with a target pathogen and adjusted for potential confounders, the excess mortality attributable to AMR BSI was 29.7 (95%CI: 26.1, 33.2) percentage points. This suggests that if all the AMR BSI were replaced by no infection, 130 (95%CI: 114, 145) deaths among 443 patients with AMR BSI might have been prevented. In conclusion, the burden of AMR infections in Indonesian hospitals is likely high. Similar large-scale evaluations should be performed across LMICs to inform interventions to mitigate AMR-associated mortality.

**Funding:** The study was funded by the Wellcome Trust (219651/Z/19/Z) to PT. For the purpose of Open Access, the author has applied a CC BY public copyright license to any Author Accepted Manuscript version arising from this submission. The funders of the study had no role in study design, data collection and analysis, decision to publish, or writing of the report.

**Competing interests:** The authors have declared that no competing interests exist.

## Introduction

According to the most comprehensive global analysis to date, antimicrobial-resistant (AMR) bacterial infections caused an estimated 4.95 million deaths associated with AMR infections, with worst impacts in low- and middle-income countries (LMICS) in 2019 [1]. The 4.95 million deaths were estimated using predictive statistical modelling and based on a counterfactual scenario of no infections (i.e. if AMR infections would have been prevented, patients would not have experienced an infection) [1]. Alternatively, 1.27 million deaths were also estimated based on an alternative counterfactual scenario of antimicrobial-susceptible (AMS) infections (i.e. if AMR infections would have been prevented, AMR infections would have been replaced by AMS infections) [1]. The study also highlighted the limited availability of data in LMICs [1]. Lim *et al* recently estimated that deaths attributable to AMR in Thailand were about 19,122 per year and highlighted the benefit of integrating information from readily available routinely collected databases in LMICs [2].

The proportion, rate and burden of AMR bacterial infections in many LMICs, including Indonesia, is largely unknown. Indonesia is a diverse lower-middle income country with a population of 274 million people in 2018. Microbiology services are underdeveloped, and data is rarely analyzed and reported. The first Indonesian submission from 20 tertiary sentinel hospitals to the WHO Global Antimicrobial Resistance and Use Surveillance System (GLASS) in 2020 [3] reported 79% third-generation cephalosporin-resistant *Klebsiella pneumoniae* (3GCRKP), 40% methicillin-resistant *Staphylococcus aureus* (MRSA), and 51% carbapenem-resistant *Acinetobacter* spp. (CRACI) in blood specimens. Very few publications have reported the proportion of AMR infections in Indonesia [4–9]. However, none of the reported proportion were categorized into that of community-origin BSI or hospital-origin BSI as proposed by the WHO GLASS [10]. In addition, rates (e.g. rate of patients with new AMR BSI per 100,000 tested patients) and mortality attributable to AMR infections have never been reported in Indonesia [3–8]. All of these parameters are crucial to monitor and estimate the burden of AMR in a country [2, 11, 12].

WHO GLASS recently published the standard guideline on how to estimate mortality attributable to AMR BSI [13]. To estimate attributable mortality, the guideline suggests to consider two scenarios, including replacement and additive scenarios [13]. In the replacement scenario, mortality is compared between patients with AMR BSI and AMS BSI by assuming that every infection caused by AMR bacteria would be replaced by an infection caused by AMS bacteria if the spread of AMR bacteria was prevented [14]. In the additive scenario, mortality is compared between patients with AMR BSI and patients without a BSI with a target pathogen. This approach assumes that AMR BSIs affect a different type of patient than AMS BSIs, and if the AMR BSI would have been prevented these patients would not have experienced a BSI. Thereby, the occurrence of AMR BSIs would add to the total number of BSIs [15, 16]. By considering both scenarios and adjusting for the influence of confounding factors, the lower and upper limit of the impact of AMR can be determined.

Here, we aim to evaluate proportions, rates, and mortality attributable to AMR infections in a tertiary-care hospital in Sulawesi, Indonesia.

## Materials and methods

### Study design and setting

We conducted a retrospective study using routinely available hospital admission and microbiology data sets in Dr. Wahidin Sudirohusodo hospital from 2015 to 2018. Dr. Wahidin Sudirohusodo Hospital was situated in the capital city of South Sulawesi province and served as a

referral hospital with 936-bed capacity. South Sulawesi covered 46,717 km$^2$ with a population of 8.7 million in 2018. Data of all hospitalized patients were electronically recorded in a hospital information system (SIMpel).

The microbiology laboratory at the study hospital used an automated blood culture system (BacT/ALERT, bioMérieux, Inc., Durham, North Carolina) since 2012. Bacterial identification and antimicrobial susceptibility testing (AST) were performed using VITEK2 (bioMérieux SA., Marcy l'Etoile, France). AST-GN93 or N317 test kits were used for Gram-negative organisms, and AST-GP67 test kits for Gram-positive organisms. The interpretation of the antibiotic disk diffusion method was based on Clinical and Laboratory Standards Institute (CLSI) guidelines.

## Data collection

Hospital admission and microbiology data were extracted separately from the hospital information system. The admission data consisted of patient hospital number, admission number, age, sex, admission date, discharge date, ward, outcome and diagnosis. The microbiology data consisted of patient hospital number, admission number, sampling date, culture identification results and AST results.

## Definitions

BSI is defined as the presence of pathogenic bacteria or fungus in the blood. The study evaluated BSI caused by *S. aureus*, *Escherichia coli*, *K. pneumoniae*, *Pseudomonas aeruginosa* or *Acinetobacter* spp., included in the 2015 WHO global priority pathogen list [17] and are of clinical importance in hospitals in the Southeast Asian region [18]. The proportion of AMR is defined as the number of AMR BSIs over all BSIs (for the pathogen of interest) [13].

A common commensal identified in a single blood specimen was considered a contaminant. Common commensal organisms include, but are not limited to, diphtheroids, *Bacillus* spp. (not *B. anthracis*), *Propionibacterium* spp., coagulase-negative staphylococci, viridans group streptococci, *Aerococcus* spp. *Micrococcus* spp. and *Rhodococcus* spp. The list of common commensals of the CDC's National Healthcare Safety Network (NHSN) was used [19]. According to the CDC's NHSN recommendation [19], a repeated blood culture positivity for a common commensal organism within the same admission is defined as a BSI of a pathogen in the study. Blood culture contamination rate is defined as the number of contaminated cultures per total number of blood cultures received by the laboratory during the study period [20].

As recommended by the WHO GLASS guideline for estimating attributable mortality of AMR BSI, we categorized patients admitted to the study hospital into patients with AMR BSI (Cohort 1), with AMS BSI (Cohort 2) and without a BSI with a target pathogen (Cohort 3) [13]. In this study, patients with AMR BSI (Cohort 1) are defined as all patients with a BSI caused by third-generation cephalosporin-resistant *E. coli* (3GCREC), 3GCR plus carbapenem-resistant *E. coli* (CREC), 3GCRKP, 3GCR plus carbapenem-resistant *K. pneumoniae* (CRKP), MRSA, CRACI, or carbapenem resistant *P. aeruginosa* (CRPA). Patients with AMS BSI (Cohort 2) are defined as all patients with a BSI caused by an AMS target pathogen (i.e. third-generation cephalosporin-susceptible *E. coli* (3GCSEC) and carbapenem susceptible *P. aeruginosa* (CSPA)). Patients without a BSI with a target pathogen (Cohort 3) was defined as all patients who did not have a BSI with a target pathogen and did not require confirmation by a negative blood culture [13].

Community-origin BSI is defined as a BSI occurring in an individual who had been admitted to a hospital for two or less calendar days, with calendar day one equal to the day of admission [13]. Hospital-origin is defined as a BSI occurring in an individual who had been

admitted to a hospital for more than two calendar days, with calendar day one equal to the day of admission [13].

Attributable mortality is defined as the excess mortality among patients with AMR BSI when compared to patients without such an infection, adjusted for the influence of confounding factors [13].

### Data analysis

Data were summarized with medians and interquartile ranges (IQR) for continuous measures, and proportions for discrete measures. IQR are presented in terms of 25th and 75th percentiles. Continuous variables and proportions were compared between groups using Kruskal-Wallis tests and chi-square tests, respectively.

The excess mortality and deaths were analyzed as recommended by the WHO GLASS [13]. First, considering the replacement scenario, we assessed the risk of mortality between patients with AMR BSI (Cohort 1) and AMS BSI (Cohort 2) using univariable and multivariable logistic regression models. Potential confounders evaluated included sex, age group, reason of admission (elective or emergency admission), direct admission to the ICUs, length of hospital stay prior to BSI, Charlson Comorbidity Index (CCI) score. An interaction term between AMR and pathogens was included in the model to evaluate the impact of AMR in different bacteria.

Second, considering the additive scenario, we assessed the risk of mortality between patients with AMR BSI (Cohort 1) and non-infection (Cohort 1, 2 and 3) using a match case-control data and conditional logistic regression models. Matched controls (1:32) were randomly selected from Cohort 1, 2 and 3 at the time of Cohort 1 having AMR BSI (using the sttocc command in STATA). This means that Cohort 1 who had later infection can be a control of another case of Cohort 1 [13]. Every patient in Cohort 1 was matched based on length of hospital stay prior to BSI, age group (neonatal [age $\leq$28 days], pediatric [age >28 days to <18 year] and adults [age $\geq$18 years]) and reason for admission on an individual level. Then, we assessed the risk of mortality between patients with AMR BSI (Cohort 1) and matched controls using univariable and multivariable conditional logistic regression models. Potential confounders evaluated, in addition to the matching variables, included sex, direct admission to the ICU and CCI score.

Excess mortality was estimated using the margins command in STATA. All analyses were performed using STATA version 14.2 (StataCorp LP, College station, Texas). Detailed AMR surveillance reporting was developed using the AutoMated tool for Antimicrobial resistance Surveillance System (AMASS) [21].

### Ethics

Study approval was obtained from the Institutional Review Board of Hasanuddin University (1062/H4.8.4.5.31/PP36-KOMETIK/2018) and the Education and Research Department of Dr. Wahidin Sudirohusodo Hospital (LB.02.01/2.2/2287/2019). Written consent was given by the director of the hospital to use their routine hospital database for research. Consent was not sought from the patients as this was a retrospective study, and the Ethical and Scientific Review Committees approved of the secondary use of routine data.

## Results

### Blood culture utilization and BSI

Of 77,752 hospitalized patients admitted to the Wahidin hospital, South Sulawesi, Indonesia from January 2015 to December 2018 (total 123,666 admissions), 8,341 patients had at least

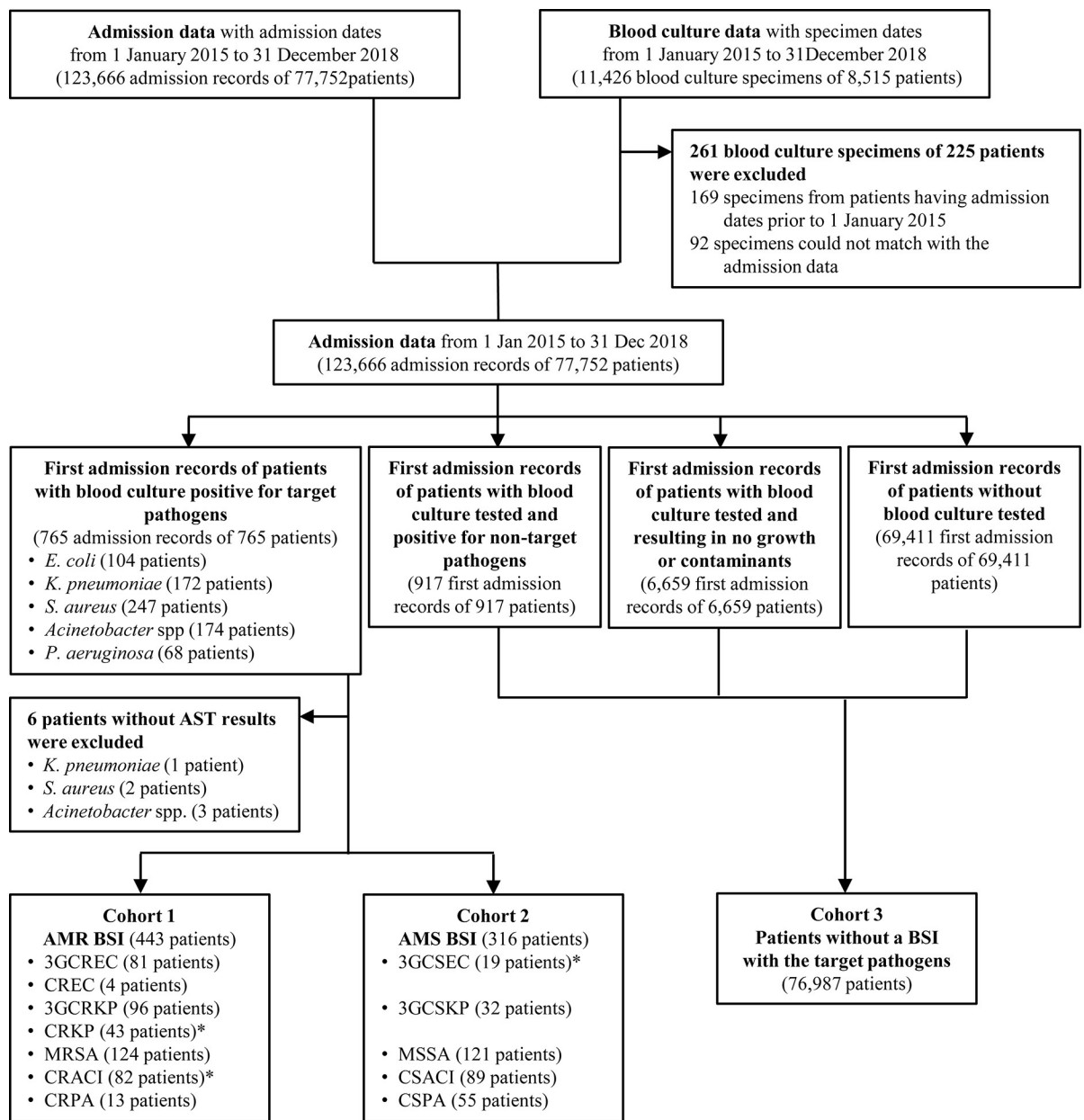

**Fig 1. Flow diagram.** For cohort 1 and cohort 2, only the first admission with BSI caused by the target pathogens was included in the analyses. For cohort 3, only the first admission was included in the analyses. Target pathogens included *E. coli* (EC), *K. pneumoniae* (KP), *S. aureus* (SA), *Acinetobacter* spp. (ACI) and *P. aeruginosa* (PA); 3GCREC = 3rd generation cephalosporin resistant and carbapenem-susceptible *E. coli*; CREC = 3rd generation cephalosporin resistant and carbapenem-resistant *E. coli*; MRSA = methicillin-resistant *S. aureus*; CRACI = carbapenem-resistant *Acinetobacter* spp.; 3GCSEC = 3rd generation cephalosporin-susceptible *E. coli*; CSACI = carbapenem-susceptible *Acinetobacter* spp.

one blood culture taken (total 11,165 blood cultures) (Fig 1). Total patient-days were 1,236,794, giving a blood culture utilization rate of 9.0 blood cultures per 1,000 patient-days. Of 8,341 patients tested for blood culture, 7,037 (84%), 1,024 (12%), 215 (3%), 38 (0.5%) patients had one, two, three, and at least four blood cultures per admissions, respectively. Among patients who had repeated blood cultures, the median time between the first and second blood culture was 8 calendar days (IQR, 4–16 days).

Of 11,165 blood cultures, 802 were regarded as contaminants, giving a blood culture contamination rate of 7.2%. The most common contaminant was coagulase-negative staphylococci (n = 646; 80.6%), followed by *Bacillus* spp (n = 51; 6.4%), *Kocuria* spp, (n = 42; 5.2%), *Aerococcus viridans*, (n = 24; 3.0%) *Micrococcus* spp, (n = 20; 2.5%) *Dermacoccus nishinomiyaensis*, (n = 16; 2%), viridans group streptococci (n = 2; 0.3%) and *Rothia dentocariosa* (n = 1; 0.1%).

Overall, 1,682 patients had at least one episode of BSI during the study period (S1 Table). The most common pathogen observed were *B. cepacia* (n = 520; 30.9%), *S. aureus* (n = 247; 14.7%), followed by *Acinetobacter* spp. (n = 174; 10.3%) and *K. pneumoniae* (n = 172: 10.2%) (S1 Table). Three patients had polymicrobial infections.

Of 765 patients having a BSI with a target pathogen, 6 had no AST results and were excluded from further analysis (Fig 1). Therefore, a total of 759 patients with first episodes of BSI caused by *E. coli* (n = 104), *K. pneumoniae* (n = 171), *Acinetobacter* spp. (n = 171), *P. aeruginosa* (n = 68) and *S. aureus* (n = 245) were included in the main analysis. Those included three patients with polymicrobial infections; a patient with BSI with 3GCRKP and carbapenem-susceptible *P. aeruginosa* (CSPA) was included as a BSI with 3GCRKP, a BSI with CRACI and *S. maltophilia* was included as a BSI with CRACI; and a BSI with 3rd generation cephalosporin-susceptible *E. coli* (3GCSEC) and *Salmonella* spp. was included as a BSI with 3GCSEC.

## Proportion and rate of AMR BSI of the target pathogens

Of 104 patients with BSI caused by *E. coli*, 81 (78%) and 4 (4%) were caused by 3GCREC and CREC, respectively (S2 Table). Of 171 patients with BSI caused by *K. pneumoniae*, 96 (56%) and 43 (25%) were caused by 3GCRKP and CRKP, respectively. The proportion of *Acinetobacter* spp. and *P. aeruginosa* BSI being caused by CRACI and CRPA was 48% (82/171) and 19% (13/68), respectively. The proportion of *S. aureus* BSI being caused by MRSA was 51% (124/245).

Stratifying BSI into BSI of community-origin or hospital-origin, we found that the proportion of 3GCREC among patients with BSI of hospital-origin was higher than that of community-origin (82% [75/92] vs. 50% [6/12], p = 0.01; S2 Table). The proportion of CRACI among patients with BSI of hospital-origin was also higher than that of community-origin (50% [81/163] vs. 13% [1/8], p = 0.04). The proportion of CREC, 3GCRKP, CRKP, MRSA and CRPA were not significantly different by infection origin.

The detailed AMR surveillance report is provided in S1 File. Among hospital-origin BSI, the highest rate of AMR BSI per 100,000 tested patients was 3GCRKP (1,764 per 100,000 tested patients), followed by MRSA (1,478 per 100,000 tested patients) and CRACI (1,282 per 100,000 tested patients) (S1 File).

## Baseline characteristics

Using the method proposed by WHO GLASS [13], we categorized the 759 patients with a BSI caused by an AMR target pathogen into Cohort 1 (AMR BSI; n = 443 patients) and by an AMS target pathogen into Cohort 2 (AMS BSI; n = 316 patients) (Fig 1). The first admission of 76,987 patients who were not included in Cohort 1 and 2 were categorized as patients without a BSI with a target pathogen (Cohort 3) and also included in the analysis.

The baseline demographics of cohort 1, 2 and 3 are shown in Table 1. Overall, the percentage of males was 53% and was not significantly different between cohorts (p = 0.42). The median age of patients with AMR BSI was lowest at 41.4 years (IQR 3.4–55.8 years), while the median age of patients with AMS BSI and Cohort 3 was 43.2 years (IQR 13.4–55.2 years) and

**Table 1. Baseline demographics.**

| Characteristic | Total* (n = 77,752 patients) | AMR BSI (n = 443 patients) | AMS BSI (n = 316 patients) | Patients without a BSI with a target pathogen (n = 76,987 patients) |
|---|---|---|---|---|
| Male sex | 41,182 (53.0%) | 247 (55.8%) | 162 (51.3%) | 40,770 (53.0%) |
| Age (years old) | | | | |
| <1 | 5,859 (7.5%) | 93 (21.0%) | 22 (7.0%) | 5,705 (7.4%) |
| 1 to <5 | 2,943 (3.8%) | 25 (5.6%) | 30 (9.5%) | 2,912 (3.8%) |
| 5 to <15 | 5,290 (6.8%) | 24 (5.4%) | 32 (10.1%) | 5,239 (6.8%) |
| 15 to <25 | 8,757 (11.3%) | 30 (6.8%) | 28 (8.9%) | 8,704 (11.3%) |
| 25 to <35 | 8,691 (11.2%) | 24 (5.4%) | 22 (7.0%) | 8,642 (11.2%) |
| 35 to <45 | 10,373 (13.3%) | 44 (9.9%) | 35 (11.1%) | 10,296 (13.4%) |
| 45 to <55 | 13,443 (17.3%) | 81 (18.3%) | 67 (21.2%) | 13,278 (17.3%) |
| 55 to <65 | 12,228 (15.7%) | 76 (17.2%) | 49 (15.5%) | 12,111 (15.7%) |
| ≥ 65 | 10,168 (13.1%) | 46 (10.4%) | 31 (9.8%) | 10,100 (13.1%) |
| Admitted from | | | | |
| Outpatient Department | 20,358 (26.2%) | 79 (17.8%) | 47 (14.9%) | 20,183 (26.2%) |
| Emergency Department | 57,394 (73.8%) | 364 (82.2%) | 269 (85.1%) | 56,804 (73.8%) |
| Direct admission to the ICU | 12,804 (16.5%) | 174 (39.3%) | 55 (17.4%) | 12,725 (16.5%) |
| Length of stay in hospital (days) ** | | | | |
| <7 | 33,489 (43.1%) | 44 (9.9%) | 32 (10.1%) | 33,188 (43.1%) |
| 7 to 14 | 28,248 (36.3%) | 95 (21.4%) | 92 (29.1%) | 27,867 (36.2%) |
| 15–30 | 12,626 (16.2%) | 160 (36.1%) | 114 (36.1%) | 12,544 (16.3%) |
| >30 | 3,376 (4.3%) | 144 (32.5%) | 78 (24.7%) | 3,375 (4.4%) |
| CCI score | | | | |
| No comorbidities (0) | 36,260 (46.6%) | 175 (39.5%) | 97 (30.7%) | 35,899 (46.6%) |
| Mild (1–2) | 19,326 (24.9%) | 83 (18.7%) | 64 (20.3%) | 19,136 (24.9%) |
| Moderate (3–4) | 12,424 (16.0%) | 72 (16.3%) | 63 (19.9%) | 12,288 (16.0%) |
| Severe (≥5) | 9,742 (12.5%) | 113 (25.5%) | 92 (29.1%) | 9,664 (12.6%) |
| In-hospital mortality | 9,835 (12.7%) | 220 (49.7%) | 144 (45.6%) | 9,925 (12.9%) |

BSI = bloodstream infections; AMR = antimicrobial resistant; AMS = antimicrobial susceptible

* AMR BSI (Cohort 1) is defined as all patients with a BSI caused by a drug-resistant target pathogen. AMS BSI (Cohort 2) is defined as all patients with a BSI caused by a drug-sensitive target pathogen. Patients without a BSI with a target pathogen (Cohort 3) is defined as all patients who did not have a BSI caused by a target pathogen and did not require confirmation by a negative blood culture. Target pathogens included *E. coli*, *K. pneumoniae*, *S. aureus*, *Acinetobacter* spp. and *P. aeruginosa*. For the cohort 1 and cohort 2, only the first admission with BSI of the first isolate was included in the analyses. For the cohort 3, only the first admission was included in the analyses

** Length of stay in hospital is defined by the duration between discharge date and admission date.

42.3 years (IQR 21.2–57.1 years) (p<0.001). The percentage of admission from the emergency department was highest among the patients with AMS BSI (85.1%; n = 269/316), followed by the patients with AMR BSI (82.2%; n = 364/443) and Cohort 3 (73.8%; n = 56,804/76,987; p<0.001). The median CCI score was higher in patients with AMR BSI (2; IQR 0–5) and AMS BSI (2; IQR 0–5) than that of Cohort 3 (1; IQR 0–3) (p<0.001). The median length of hospital stay prior to the diagnosis of BSI was higher among patients with AMR BSI than that of AMS BSI (9 days [IQR 3–19 days] vs 8 days [IQR 3–14 days], p = 0.04).

In the cohort 3, most patients (90.2%, n = 69,411/76,987) had no blood culture taken, 8.6% of patients (6,659/76,897) had blood culture taken and culture negative for pathogens, and 1.2% of patients (917/76,897) had blood culture positive for a non-target pathogen.

### In-hospital mortality

Of overall, in-hospital mortality was 12.7% (9,835/77,752). The in-hospital mortality was highest among patients with AMR BSI (49.7%; n = 220/443) followed by among patients with AMS BSI (45.6%; n = 144/316) and Cohort 3 (12.9%; n = 9,925/76,986) (p<0.001). The median total duration of hospital stay was highest among patients with AMR BSI (22; IQR 13–35) followed by patients with AMS BSI (17; IQR 10–30) and Cohort 3 (7; IQR 4–13) (p<0.001; Table 1).

Using univariable logistic regression to evaluate factors associated with mortality among patients with AMR-BSI and AMS-BSI (Table 2), we found that age, admission from the Emergency Department, longer length of hospital stays prior to the diagnosis of BSI and pathogens were associated with mortality. Although sex and direct admission to the ICUs were not strongly associated with mortality (all p>0.10), we included them in the final multivariable models together with age, type of admission (admitted from outpatient department or emergency department), length of hospital stays prior to the diagnosis of BSI, CCI score and pathogens as they are predefined potential confounders for mortality. In the final multivariable logistic regression model, patient with CRACI BSI (adjusted OR 4.10, 95%CI: 1.13–14.92), patients with the length of stay prior to the diagnosis of BSI ≥30 days (adjusted OR 2.17, 95% CI: 1.16–4.06), patients with CRPA BSI (adjusted OR 2.08, 95%CI: 0.40–10.92) and patients admitted from the emergency department (adjusted OR 1.99, 95%CI 1.30–3.04) were the strongest independent risk factors associated with mortality (Table 2).

Using univariable conditional logistic regression on a matched case-control data set to evaluate factors associated with mortality among patients with AMR-BSI and those without a BSI with the target pathogens (Table 3), we found that sex, age, direct admission to the ICUs, CCI score and AMR infections were strongly associated with mortality. In the final multivariable conditional logistic regression model, sex, age, direct admission to the ICU, CCI score and AMR infections were independently associated with mortality. Patients with AMR BSI had about 391% higher risk of death than those without a BSI with a target pathogen (adjusted conditional OR 4.91, 95%CI: 4.02–6.00, p<0.001, Table 3).

### Excess mortality and deaths caused by AMR BSI

Compared with patients with AMS BSI and adjusted for potential confounders, the estimated excess mortality attributable to AMR infection was not statistically different (Table 4). However, compared with patients without a BSI with a target pathogen and adjusted for potential confounders, the estimated excess mortality attributable to AMR infections was 29.7 (95% CI:26.1, 33.2) percentage points. This suggests that 130 (95%CI:114, 145) deaths among 443 patients with AMR BSI might have been prevented if all the AMR BSI in this study did not occur (Table 4).

### Discussion

Our retrospective study has demonstrated that AMR surveillance report and mortality attributable to AMR in hospitals in LMICs could be assessed by integrating information from readily available patient databases. Our data show that the proportion of AMR among patients with BSI of both community-origin and hospital-origin at the study hospital is high. For example, the proportion of community-origin 3GCREC was 50% and the proportion of hospital-origin 3GCREC was 82%. Compared to patients with AMS BSI and adjusted for potential confounders, the mortality of patients with AMR BSI was not significantly different. However, compared with patients without a BSI with the target pathogens and adjusted for potential confounders, the mortality of patients with AMR BSI was estimated to be 29.7 percentage points higher. The burden of AMR BSI at the study hospital is high with excess deaths of 130

**Table 2. Factors associated with in-hospital mortality in patients with AMR BSI compared with AMS BSI.**

| Factor | In-hospital mortality in Cohort 1 (n = 443 patients) | In-hospital mortality in Cohort 2 (n = 316 patients) | Crude ORs (95%CI) | P value | Adjusted ORs (95%CI) | P value |
|---|---|---|---|---|---|---|
| Sex | | | | | | |
| Female | 53.6% (105/196) | 44.8% (69/154) | 1.0 | 0.37 | 1.0 | 0.49 |
| Male | 46.6% (115/247) | 46.3% (75/162) | 0.88 (0.66–1.17) | | 0.90 (0.66–1.22) | |
| Age (years old) | | | | | | |
| <1 | 48.4% (45/93) | 40.9% (9/22) | 0.82 (0.50–1.33) | | 0.85 (0.44–1.63) | |
| 1 to <5 | 32.0% (8/25) | 36.7% (11/30) | 0.49 (0.26–0.93) | | 0.44 (0.22–0.89) | |
| 5 to <15 | 50.0% (12/24) | 40.6% (13/32) | 0.74 (0.40–1.38) | | 0.71 (0.36–1.39) | |
| 15 to <25 | 53.3% (16/30) | 42.9% (12/28) | 0.86 (0.47–1.58) | | 0.73 (0.38–1.41) | |
| 25 to <35 | 33.3% (8/24) | 40.9% (9/22) | 0.54 (0.27–1.07) | | 0.53 (0.26–1.10) | |
| 35 to <45 | 40.9% (18/44) | 40.0% (14/35) | 0.63 (0.36–1.09) | | 0.60 (0.33–1.08) | |
| 45 to <55 | 51.9% (42/81) | 52.2% (35/67) | 1.0 | 0.01 | 1.0 | 0.06 |
| 55 to <65 | 52.6% (40/76) | 42.9% (21/49) | 0.88 (0.55–1.41) | | 0.87 (0.52–1.43) | |
| ≥ 65 | 67.4% (31/46) | 64.5% (20/31) | 1.81 (1.02–3.20) | | 1.55 (0.85–2.84) | |
| Admitted from | | | | | | |
| Outpatient Department | 41.8% (33/79) | 27.7% (13/47) | 1.0 | 0.005 | 1.0 | 0.001 |
| Emergency Department | 51.4% (187/364) | 48.7% (131/269) | 1.76 (1.18–2.61) | | 1.99 (1.30–3.04) | |
| Direct admission to the ICU | 51.7% (90/174) | 52.7% (29/55) | 1.26 (0.92–1.72) | 0.15 | 1.28 (0.85–1.93) | 0.24 |
| Length of stay prior to the diagnosis of BSI (calendar day) * | | | | | | |
| 1 to 2 | 40.0% (26/65) | 45.2% (19/42) | 1.0 | 0.01 | 1.0 | 0.05 |
| 3 to 7 | 45.2% (61/135) | 41.4% (46/111) | 1.06 (0.67–1.68) | | 0.95 (0.59–1.55) | |
| 8 to 14 | 42.7% (38/89) | 54.4% (49/90) | 1.30 (0.80–2.11) | | 1.15 (0.68–1.96) | |
| 15–30 | 58.1% (54/93) | 31.9% (15/47) | 1.34 (0.81–2.22) | | 1.16 (0.67–2.03) | |
| >30 | 67.2 (41/61) | 57.7% (15/26) | 2.49 (1.39–4.46) | | 2.17 (1.16–4.06) | |
| CCI score | | | | | | |
| No comorbidities (0) | 44.6% (78/175) | 41.2% (40/97) | 1.0 | 0.09 | 1.0 | 0.23 |
| Mild (1–2) | 48.2% (40/83) | 40.6% (26/64) | 1.06 (0.71–1.59) | | 0.89 (0.54–1.47) | |
| Moderate (3–4) | 56.9% (41/72) | 44.4% (28/63) | 1.36 (0.90–2.06) | | 1.31 (0.77–2.25) | |
| Severe (≥5) | 54.0% (61/113) | 54.4% (50/92) | 1.54 (1.07–2.21) | | 1.37 (0.85–2.20) | |
| Organisms ** | | | | | | |
| AMS *E. coli* | - | 52.6% (10/19) | 1.0 | 0.01 | 1.0 | 0.04 |
| AMR *E. coli* | 51.8% (44/85) | - | 0.97 (0.36–2.61) | | 0.87 (0.30–2.49) | |

*(Continued)*

**Table 2.** (Continued)

| Factor | In-hospital mortality in Cohort 1 (n = 443 patients) | In-hospital mortality in Cohort 2 (n = 316 patients) | Crude ORs (95%CI) | P value | Adjusted ORs (95%CI) | P value |
|---|---|---|---|---|---|---|
| AMS *K. pneumoniae* | - | 53.1% (17/32) | 1.02 (0.33–3.18) | | 1.15 (0.35–3.75) | |
| AMR *K. pneumoniae* | 46.0% (64/139) | - | 0.77 (0.29–2.01) | | 1.74 (0.49–6.25) | |
| AMS *S. aureus* | - | 39.7% (48/121) | 0.59 (0.22–1.56) | | 0.64 (0.23–1.77) | |
| AMR *S. aureus* | 40.3% (50/124) | - | 0.61 (0.23–1.60) | | 1.32 (0.36–4.91) | |
| AMS *Acinetobacter* spp. | - | 44.9% (40/89) | 0.73 (0.27–1.98) | | 0.75 (0.26–2.14) | |
| AMR *Acinetobacter* spp. | 67.1% (55/82) | - | 1.83 (0.67–5.04) | | 4.10 (1.13–14.92) | |
| AMS *P. aeruginosa* | - | 52.7% (29/55) | 1.00 (0.35–2.85) | | 1.20 (0.40–3.61) | |
| AMR *P. aeruginosa* | 53.9% (7/13) | - | 1.05 (0.26–4.32) | | 2.08 (0.40–10.92) | |

BSI = bloodstream infections; AMR = antimicrobial resistant; AMS = antimicrobial susceptible; CCI = Charlson Comorbidity Index

* Time of diagnosis is defined as the date positive blood culture was taken. Community-origin BSI is defined as a confirmed BSI occurring in an individual who has been admitted to a hospital for two or less calendar days, with calendar day one equal to the day of admission. Hospital-origin BSI is defined as a confirmed BSI occurring in an individual who has been admitted to a hospital for more than two calendar days, with calendar day one equal to the day of admission. Community-origin or hospital-origin BSI was taken account by the length of stay prior to the diagnosis of BSI

** Interaction term between AMR and organism was included in the model. The interaction term was used to evaluate the impact of AMR in different bacteria.

deaths. Specifically, 130 deaths among 443 patients with AMR BSI might have been prevented if all the AMR BSI in this study were replaced with no infection.

The high burden of AMR in LMICs when compared with patients without infection is consistent with the recent report by the Antimicrobial Resistance Collaborators [1] and other studies [22–25]. Based on the counterfactual of no infection, the Antimicrobial Resistance Collaborators estimated that 4.95 million (3.62–6.57 million) deaths are associated with bacterial AMR infections in 2019, and that the highest burden is in South Asia (1.39 million deaths), sub-Saharan Africa (1.07 million deaths), and Southeast Asia, east Asia, and Oceania, (1.02 million deaths) [1]. *E. coli*, *S. aureus*, *K. pneumoniae*, *A. baumannii*, and *P. aeruginosa* are all leading pathogens associated with deaths associated with AMR infections worldwide [1].

Our findings support the critical need of improving AMR surveillance system, reducing the rate of AMR infection (per 100,000 patients per year and per 100,000 tested patients) with better infection prevention and control measures, and optimizing use of antimicrobials in hospitals in LMICs [1], as also proposed by the national action plans for AMR Indonesia [26]. The high rate (per 100,000 tested patients) of hospital-origin MRSA BSI observed in our study also highlights the need to evaluate and improve hand hygiene practice, which is a fundamental part of an infection prevention and control strategy, in the hospital [27].

In our study we did not find a difference in risk of mortality between patients with AMR BSI and AMS BSI; this finding should be considered with caution and could be due to several reasons. First, carbapenems were commonly used as the first or second empirical antibiotic at the study hospital. This could mitigate any risk differences between patients with AMR *E. coli* BSI compared with AMS *E. coli* BSI (aOR 0.87) where most of AMR *E. coli* is 3GCREC. However, the overuse of carbapenems could also result in the high proportions and rates of CRKP, CRACI and CRPA infections in the hospital. Patients with carbapenem-resistant infections

**Table 3. Factors associated with in-hospital mortality in patients with AMR BSI compared with patients without a BSI with a target pathogen.**

| Factors | In-hospital mortality in Cohort 1 (n = 443 patients) | In-hospital mortality in Cohort 3 (n = 76,987 patients) | Crude cORs* (95%CI) | P value | Adjusted cORs* (95%CI) | P value |
|---|---|---|---|---|---|---|
| Sex | | | | | | |
| Female | 53.6% (105/196) | 12.0% (4,360/36,217) | 1.0 | | 1.0 | |
| Male | 46.6% (115/247) | 13.7% (5,565/40,769) | 0.84 (0.77–0.92) | <0.001 | 0.85 (0.78–0.93) | <0.001 |
| Age (years old) | | | | | | |
| <1 | 48.4% (45/93) | 14.9% (847/5,705) | 0.84 (0.55–1.27) | | 0.95 (0.61–1.47) | |
| 1 to <5 | 32.0% (8/25) | 9.3% (270/2,912) | 0.44 (0.29–0.68) | | 0.48 (0.31–0.74) | |
| 5 to <15 | 50.0% (12/24) | 8.3% (436/5,239) | 0.45 (0.30–0.67) | | 0.51 (0.34–0.77) | |
| 15 to <25 | 53.3% (16/30) | 8.6% (746/8,704) | 0.51 (0.40–0.65) | | 0.64 (0.50–0.83) | |
| 25 to <35 | 33.3% (8/24) | 9.8% (850/8,642) | 0.62 (0.51–0.76) | | 0.73 (0.59–0.90) | |
| 35 to <45 | 40.9% (18/44) | 11.5% (1,185/10,296) | 0.78 (0.66–0.93) | | 0.90 (0.75–1.07) | |
| 45 to <55 | 51.9% (42/81) | 14.4% (1,907/13,278) | 1.0 | <0.001 | 1.0 | <0.001 |
| 55 to <65 | 52.6% (40/76) | 15.3% (1,847/12,110) | 0.85 (0.73–1.00) | | 0.76 (0.65–0.90) | |
| ≥65 | 67.4% (31/46) | 18.2% (1,837/10,100) | 0.99 (0.84–1.17) | | 0.81 (0.68–0.96) | |
| Direct admission to the ICU | 51.7% (90/174) | 21.0% (2,671/12,725) | 1.49 (1.33–1.66) | <0.001 | 1.53 (1.36–1.72) | <0.001 |
| CCI score | | | | | | |
| No comorbidities (0) | 44.6% (78/175) | 8.5% (3,032/35,899) | 1.0 | <0.001 | 1.0 | <0.001 |
| Mild (1–2) | 48.2% (40/83) | 13.2% (2,528/19,135) | 1.42 (1.23–1.64) | | 1.41 (1.20–1.66) | |
| Moderate (3–4) | 56.9% (41/72) | 17.6% (2,168/12,288) | 1.77 (1.52–2.06) | | 1.82 (1.52–2.17) | |
| Severe (≥5) | 54.0% (61/113) | 22.7% (2,197/9,664) | 2.26 (1.98–2.59) | | 2.33 (2.01–2.71) | |
| Type of infections | | | | | | |
| No infections | - | 12.9% (9,925/76,986) | 1.0 | <0.001 | 1.0 | <0.001 |
| AMR infections | 49.7% (220/443) | - | 5.30 (4.36–6.45) | | 4.91 (4.02–6.00) | |

BSI = bloodstream infections; AMR = antimicrobial resistant; CCI = Charlson Comorbidity Index

* Conditional odds ratios (cOR) for in-hospital mortality were estimated from a conditional logistic regression and a match case-control data. Matched controls (1:32) were randomly selected from Cohort 1, 2 and 3 at the time of Cohort 1 having AMR infection. This means that Cohort 1 who had later infection can be a control of another case of Cohort 1. Every patient in Cohort 1 was matched based on duration of hospital stay prior to the infection, age group (neonatal, pediatric, adult), and reason for admission (elective or emergency admission) on an individual level.

have a high risk of mortality, and, overall, overuse of carbapenem could lead to a higher burden of AMR as shown in the additive scenario analysis. Second, empirical antibiotics are commonly prescribed without taking blood cultures at the study hospital, because of restricted reimbursement under the National Health Insurance System (NHIS; or Jaminan Kesehatan

**Table 4. Excess mortality and deaths among patients with AMR BSI.**

| Models | Excess mortality (95% CI)* | Excess deaths (95% CI)** |
|---|---|---|
| AMR BSI vs. AMS BSI * | -0.01 percentage points (-15.4 to 15.4 percentage points) | 0 deaths (-34 to 34 deaths) |
| AMR BSI vs. Non-infection ** | 29.7 percentage points (26.1 to 33.2 percentage points) | 130 deaths (114 to 145 deaths) |

BSI = bloodstream infections; AMR = antimicrobial resistant; AMS = antimicrobial susceptible

* Adjusted for sex, age group, reason of admission, ICU admission, length of stay prior to BSI, Charlson Comorbidity Index (CCI) score and pathogens

** Adjusted for sex, age group, ICU admission and CCI score.

Nasional) [28]. It is still unclear how low blood culture utilization rate or delayed blood culture could impact the comparison of mortality between patients with AMR BSI and AMS BSI, and between patients with AMR BSI and non-infection [29]. Nonetheless, we hypothesized that a number of patients with AMR BSI died without being diagnosed because of the low blood culture utilization rate; therefore, excess deaths attributable to AMR BSI was likely underestimated. Third, our study could lack power to differentiate a difference in risk of mortality between patients with AMR BSI and AMS BSI. Our previous study including 9,796 BSI patients from 10 provincial hospitals in Thailand found a higher mortality of patients with multidrug-resistant (MDR) BSI compared with non-MDR BSI; the mortality attributable to MDR was 7% in community-acquired bacteremia, 15% in healthcare-associated bacteremia and 15% in hospital-acquired bacteremia [2]. The study had a much higher sample size [2] and a much higher blood culture utilization rate [30]. Fourth, it could be due to residual confounding factors.

Analyzing local data can additionally unveil potential issues, which require further local evaluation and actions. For example, the local issues raised by the study include the relatively high blood culture contamination rate (7.2%) and high prevalence of *B. cepacia* as a pathogen causing BSI at the study hospital. The American Society for Microbiology (ASM) and the CLSI recommend that the overall blood culture contamination rates should not exceed 3% and the reported contamination rates in hospitals vary widely ranging from 0.6% to 12.5% [20]. The high contamination rate in our setting could be caused by poor aseptic techniques of some healthcare workers who collect the blood culture samples [20]. We consulted our findings with the infection prevention control team and physicians of the study hospital. As a result, the study hospital is planning to improve practices to reduce blood culture contamination [20], review cases of *B. cepacia* BSI, and develop a measure to reduce the burden of all hospital-acquired infections.

Our study has multiple strengths. First, we used data from readily available databases of microbiology laboratory and hospital admission; therefore, we could estimate mortality attributable to AMR BSI while adjusting for potential confounders including age, sex, CCI score and length of hospital stay prior to the diagnosis of BSI. This allows other hospitals in LMICs with readily available data to follow the similar analysis. Second, we followed the latest WHO guideline to estimate mortality, using both replacement and additive scenarios [13]. This allows us to estimate both lower and upper limit of potential impact of AMR BSI. In the additive scenario, we also randomly selected control patients using exposure density sampling as recommended by the WHO GLASS [13]. This allows us to reduce a potential bias caused by length of hospital stay prior to the diagnosis of BSI effectively. Third, we used exposure density sampling as recommended by the WHO GLASS [13], which means that for every patient with an AMR BSI by a target pathogen, an unexposed patient is matched based on duration of hospital stay before infection, on an individual level. This approach can reduce immortal time bias, which is an artificial survival advantage among patients with hospital-acquired infections who had to survive long enough to be able to develop the infection. If the exposure density sampling was not used, the impact of AMR BSI could be underestimated [13]. Fourth, we evaluated five bacteria species included in in the 2015 global priority list of AMR bacteria from the WHO [17], and used an interaction term to take account of potentially different impact of AMR in different bacteria. This allows us to estimate the overall impact of AMR BSI and observe potentially higher impact caused by CRACI and CRPA than that caused by 3GCREC and 3GCRKP at the study hospital.

Our study has several limitations. First, the magnitude of impact shown may not be generalizable to other hospitals, settings or countries. Second, the definition of origin of infection is only a proxy for community-acquired and hospital-acquired infection. Our electronic data

could not define whether patients were transferred from another hospital and duration of hospitalization at the transferring hospital. Therefore, a proportion of BSI of community-origin in our study could be hospital-origin at transferring hospitals. Third, a small proportion of patients with AMS BSI (Cohort 2) and patients without a BSI with a target pathogen (Cohort 3) might have AMR infections in other organs (such as pneumonia or urinary tract infections). Fourth, we could not determine whether the blood culture utilization at the study hospital is low because of the case mix of patients presenting at the study hospital [30] or because of patients being treated with empirical treatment without blood culture sampling [29]. Further studies utilizing antibiotic prescription data and estimating a proportion of patients having a blood culture taken within ±1 calendar day of the day when a parenteral antibiotic was started at the study hospital and continued for at least four consecutive days [29] could be helpful. Fifth, the high proportion of AMR observed in our study (e.g. high proportion of 3GCREC and high proportion of MRSA) should not be misinterpreted to recommend the use of watch antibiotics (e.g. carbapenem and vancomycin) or reserve antibiotics (e.g. colistin or tigecycline) as first-line empirical treatment at the study hospital [31]. This is because a considerable fraction of AMR reported here is probably based on blood culture after failure of the first or second empirical treatment [29]. Further study is needed to estimate the proportion of AMR prior to the first empirical treatment so that it can be used to guide choice of the first empirical therapy [29].

## Conclusion

Our study shows that careful evaluation of readily available routine databases can provide useful information on the proportion, rate and excess mortality attributable to AMR infections in hospitals in LMICs. The methodology used in our study could be applied to other geographical areas where microbiological facilities and electronical databases are readily available to provide a more comprehensive global picture of the importance of AMR infections as a cause of death.

## Supporting information

**S1 Table. Pathogenic organisms isolated from blood culture among patients presenting at Wahidin Hospital, Makassar, Indonesia, from 2015 to 2018.**
(PDF)

**S2 Table. Proportions of patients with blood cultures positive for antibiotic-resistant isolates of the five targeted pathogens.**
(PDF)

**S1 File. Antimicrobial Resistance (AMR) surveillance report.**
(PDF)

## Acknowledgments

We gratefully acknowledge the support provided by staff Dr. Wahidin Sudirohusodo Hospital, South Sulawesi, Indonesia.

## Author Contributions

**Data curation:** Patricia M. Tauran, Direk Limmathurotsakul.

**Formal analysis:** Patricia M. Tauran.

**Methodology:** Patricia M. Tauran.

**Supervision:** Niholas P. J. Day, Mansyur Arif, Direk Limmathurotsakul.

**Writing – original draft:** Patricia M. Tauran, Direk Limmathurotsakul.

**Writing – review & editing:** Patricia M. Tauran, Irawaty Djaharuddin, Uleng Bahrun, Asvin Nurulita, Sudirman Katu, Faisal Muchtar, Ninny Meutia Pelupessy, Raph L. Hamers, Direk Limmathurotsakul.

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
