## [Decision Letter · Decision Letter 0]

30 May 2022

PGPH-D-22-00421

Epidemiology and excess mortality attributable to antimicrobial-resistant bacterial infection at a tertiary-care hospital in Indonesia.

Dear Dr. Tauran,

Thank you for submitting your manuscript to PLOS Global Public Health. After careful consideration, we feel that it has merit but does not fully meet PLOS Global Public Health’s publication criteria as it currently stands. Therefore, we invite you to submit a revised version of the manuscript that addresses the points raised during the review process.

Please submit your revised manuscript by . If you will need more time than this to complete your revisions, please reply to this message or contact the journal office at globalpubhealth@plos.org. Please include the following items when submitting your revised manuscript:

We look forward to receiving your revised manuscript.

Kind regards,

Thomas P. Van Boeckel

Academic Editor

Journal Requirements:

- State the initials, alongside each funding source, of each author to receive each grant.

2. We do not publish any copyright or trademark symbols that usually accompany proprietary names, eg (R), (C), or TM  (e.g. next to drug or reagent names). Please remove all instances of trademark/copyright symbols throughout the text, including VITEK®2 on page 6.

Additional Editor Comments (if provided):

Please could you address the comments raised by referee #1 and #2

Reviewers' comments:

Reviewer's Responses to Questions

**Comments to the Author**

1. Does this manuscript meet PLOS Global Public Health’s publication criteria? Is the manuscript technically sound, and do the data support the conclusions? The manuscript must describe methodologically and ethically rigorous research with conclusions that are appropriately drawn based on the data presented.

Reviewer #1: Yes

Reviewer #2: Yes

2. Has the statistical analysis been performed appropriately and rigorously?

Reviewer #1: Yes

Reviewer #2: Yes

3. Have the authors made all data underlying the findings in their manuscript fully available (please refer to the Data Availability Statement at the start of the manuscript PDF file)?

Reviewer #1: No

Reviewer #2: No

4. Is the manuscript presented in an intelligible fashion and written in standard English?

Reviewer #1: No

Reviewer #2: Yes

5. Review Comments to the Author

Reviewer #1: In table 2, how can the authors calculate the mortality of outpatients?

In table 3, those factors were demography characterized, not epidemiology? There were two aims of this study - epidemiology, and mortality attributable, however, the authors have just described mortality attributable, not mention the epidemiology.

Reviewer #2: In their manuscript entitled, “Epidemiology and excess mortality attributable to antimicrobial-resistant bacterial infection at a tertiary-care hospital in Indonesia,” Dr. Tauran, and colleagues, utilize routinely collected admission and microbiology data from a tertiary hospital in Indonesia to explore attributable mortality of drug-resistant bloodstream infections occurring in hospitalized patients over a 3-year period (2015-18). The manuscript is well written, comprehensive in its approach and aligns well with efforts underway to address the problem of mortality due to AMR infections globally. The analytic approach to understanding excess mortality and deaths was clearly described and can be reproduced in other similar settings. Some clarity is required in terms of potential areas of bias introduced by the limitations in the datasets. The following comments are included to help address some of these minor issues:

Title: Given that the study includes only blood-stream infections, the title should explicitly state that the infections described in this study are AMR bacterial blood-stream infections rather than generally as ‘infection’.

Introduction:

Line 57: The sentence starting, “The 4.95 million deaths…” is difficult to digest and would benefit from rephrasing for clarity, perhaps alluding to similar language to describe the replacement and additive scenarios later in the manuscript.

Line 70: Rephrase for clarity, perhaps by deleting ‘which’ so that ‘reported’ can serve as the verb in the sentence.

Line 75: the definition for ‘incidence rates’ in parentheses does not actually define a ‘rate’. Suggest removing the word rate and leaving as ‘incidence’ or ‘cumulative incidence’ throughout the text.

Definitions:

Lined 135-6: Is it possible that patients included in Cohort 3 had a diagnosis for an infection caused by an AMR bacteria isolated from a compartment other than the bloodstream? If yes, then a proportion of these patients might still have a drug-resistant infection, just not a BSI. Ideally, it would be useful to understand the difference in outcome for patients with drug resistant infection that are BSI versus not BSI as a separate cohort. Were these data reported as per GLASS requirements? It is important to highlight this distinction here and elsewhere in the document (see comment above re: Title), including the limitations section in the Discussion.

Data analysis:

Lines 168-9: The sentence states that you are comparing mortality risk between patients with AMR BSI and patients without infection. Yet, the parentheses after ‘non-infection’ include Cohorts 1 and 2. Given that Cohorts 1 and 2 comprise patients with infection, further explanation is required to explain how Cohorts 1 and 2 can be used to represent patients without infection. Is it because patients who have yet to have a BSI identified by blood culture within cohorts 1 and are not yet classified as having infection? If this is the case, please also explain to what extent this approach introduces bias.

Results:

Line 223: I think the authors did not mean to include ‘Of’ at the beginning of the sentence. Please delete.

Lines 223-6: The sentence states that the most common observed pathogens were B. cepacia, S. aureus, Acinetobacter spp and K. pneumoniae. These represent a total of 1,113 (66.2%) of the total 1682 isolated pathogenic organisms described in the S1 Table. The paragraph, however, refers to 8341 patients with one episode of BSI. Please reconcile these numbers.

Lines 230-235: This information is included verbatim in the caption for Figure 1. Is it possible to remove it in one of the places to avoid redundancy?

Discussion:

Lines 387-89:

o Given the low utilization of blood cultures overall, do the authors have any data to evaluate the potential association between illness severity and clinician threshold for obtaining a blood culture (i.e., to ascertain level of selection bias)? For instance, if clinicians are more likely to order a blood culture when a patient is critically ill, then the cohorts comprising patients with AMR and AMS infections would also reflect the sickest patients and would explain why there is a significant difference in mortality between those patients and the patients in cohort 3.

o Also, do the authors know what proportion of cohort 3 patients had blood cultures but were negative? If there was a sizeable proportion of such patients, presenting these data explicitly could help allay the above concern about selection bias.

6. PLOS authors have the option to publish the peer review history of their article (what does this mean?). If published, this will include your full peer review and any attached files.

**Do you want your identity to be public for this peer review?** For information about this choice, including consent withdrawal, please see our Privacy Policy.

Reviewer #1: **Yes: **Diep The Tai

Reviewer #2: No

---

## [Editor Report · Decision Letter 1]

30 Jun 2022

Excess mortality attributable to antimicrobial-resistant bacterial bloodstream infection at a tertiary-care hospital in Indonesia

PGPH-D-22-00421R1

Dear Tauran,

We are pleased to inform you that your manuscript 'Excess mortality attributable to antimicrobial-resistant bacterial bloodstream infection at a tertiary-care hospital in Indonesia' has been provisionally accepted for publication in PLOS Global Public Health.

Best regards,

Thomas P. Van Boeckel

Academic Editor